# Water Effective Diffusion Coefficient in Dairy Powder Calculated by Digital Image Processing and through Machine Learning Algorithms of CLSM Micrographs

**DOI:** 10.3390/foods13010094

**Published:** 2023-12-27

**Authors:** Valentyn A. Maidannyk, Yuriy Simonov, Noel A. McCarthy, Quang Tri Ho

**Affiliations:** 1Food Chemistry & Technology Department, Teagasc Food Research Centre, Moorepark, Fermoy, P61 C996 County Cork, Ireland; noel.mccarthy@teagasc.ie (N.A.M.); quangtri.hqt@gmail.com (Q.T.H.); 2Independent Researcher, 6511 Nijmegen, The Netherlands; simonovyup@gmail.com; 3Institute of Marine Research, 5003–5268 Bergen, Norway

**Keywords:** diffusion, powders, Matlab, confocal microscopy, Python, machine learning

## Abstract

Rehydration of dairy powders is a complex and essential process. A relatively new quantitative mechanism for monitoring powders’ rehydration process uses the effective diffusion coefficient. This research focused on modifying a previously used labor-intensive method that will be able to automatically measure the real-time water diffusion coefficient in dairy powders based on confocal microscopy techniques. Furthermore, morphological characteristics and local hydration of individual particles were identified using an imaging analysis procedure written in Matlab©—R2023b and image analysis through machine learning algorithms written in Python™-3.11. The first model includes segmentation into binary images and labeling particles during water diffusion. The second model includes the expansion of data set selection, neural network training and particle markup. For both models, the effective diffusion follows Fick’s second law for spherical geometry. The effective diffusion coefficient on each particle was computed from the dye intensity during the rehydration process. The results showed that effective diffusion coefficients for water increased linearly with increasing powder particle size and are in agreement with previously used methods. In summary, the models provide two independent machine measurements of effective diffusion coefficient based on the same set of micrographs and may be useful in a wide variety of high-protein powders.

## 1. Introduction

The control of the hydration process of high-protein powders is important and requires a deep understanding of the physicochemical basis. Poor hydration of high-protein powders generates sediments, lumps and flecks in reconstituted systems. The rehydration process is complex and generally may be divided into four steps: wetting, sinking, dispersion and full dissolution [1]. Poor solubility and hydration properties are associated with the presence of high amounts of hydrophobic (protein–protein) interactions, which occur during manufacturing and storage [2]. A hydration mechanism is still not well understood and needs further investigation on micro and nanoscales.

Effective water diffusion coefficient *D_eff_* is an essential parameter interpreted as a quantitative representation of the hydration process. Water diffusion behavior in polymers controlled by water migration concentration can be described by Fick’s law [3]. The recently presented Confocal Laser Scanning Microscopy (CLSM) method shows the way of real-time determination of *D_eff_* and has been successfully applied for different dairy powders [2,4,5,6,7]. This method records the dynamic diffusion process and provides a set of pictures before, in time and after water penetration inside the individual powder particles. However, this method is very time-consuming, requires difficult manual measurement of particle sizes and needs further development. Also, in several cases (low intensity of the object, strong background, etc.), the raw images obtained from the CLSM method need improvement.

Furthermore, the image analysis technique may significantly simplify the determination of morphological characteristics and hydration of individual powder particles. Such a large data set requires a comprehensive set of reference-standard algorithms and workflow applications for image processing and analysis, including segmentation, noise reduction, and geometrical characteristic computation. In such cases, automatic image processing procedures performed in engineering software, i.e., Image Processing Toolbox in Matlab, can be used to reduce time consumption. At the same time, the Deep and Machine Learning approaches written in Python in image analysis may be even more precise and effective. Application of image processing procedures for determining particle size distributions, morphological operations, filtering and 2D diffusion calculation in biological objects have been reported [8,9,10,11]. At the same time, Deep and Machine Learning approach was used to distinguish polysaccharides in raspberry powders [12], composition analysis of powder mixtures [13] and for predicting fishiness off-flavor and identifying compounds of lipid oxidation in dairy powders [14]. However, these approaches have not been extended to analyze water diffusion measurement in high-protein dairy powders.

This study’s main purpose was to analyze water penetration inside particles during rehydration. To this end, particles of milk powders during the rehydration process recorded in CLSM micrographs were subjected to two independent image analysis approaches coupled with the diffusion model to compute its diffusion coefficients. The results were compared with the existing model. Milk Protein Concentrate powder was chosen as a model high-protein system. Analysis workflow was performed by Image Processing Toolbox in Matlab© and Python™. The applications of this approach include different food and dairy powders as well as pharmaceutical materials.

## 2. Materials and Methods

### 2.1. Materials

Nitrogen gas-injected spray-dried milk protein concentrate (MPC) (80%, *w*/*w*, protein) was supplied by an Irish dairy ingredient manufacturer [2].

### 2.2. Confocal Laser Scanning Microscopy (CLSM)

A Leica TCS SP5 confocal laser scanning microscope (Leica Microsystems CMS GmbH, Wetzlar, Germany) was used for visualization of particle rehydration [4]. A 57% power of DPSS 561 nm laser was used for excitation. MPC powder particles were thinly sprinkled onto a glass slide, and approximately, 50 μL of the dye mixture was added and a coverslip was placed on top. The dye mixture comprised 0.1% (wt/wt) Rhodamine B (Sigma-Aldrich Co., St. Louise, MO, USA) in milli-Q water. Eppendorf tubes (2 mL) (Eppendorf, Hamburg, Germany) containing the mixtures were vortex mixed for 30 s with IKA^®®^ MS1 Minishaker (Staufen, Germany) at 2500 min^−1^, before the mixtures were added to the dry sample on a microscope slide. The CLSM images of each experiment were taken using HCX PL APO lambda blue 63.0× oil immersion objectives with numerical apertures of 1.4 and a reflective index of 1.52. z-Stacks were obtained in order to identify the maximum diameter for diffusion measurements. The Scan speed was 400 Hz, the resolution was 440 × 440 nm Greyscale 8-bit images (512 × 512 pixels, digital zoom factors ×1) were acquired manually at approximately 3 s time intervals and were pseudo-colored green.

### 2.3. Real-Time Visualization of Liquid Phase Water Diffusion in Milk Protein Concentrate Powder

Real-time visualization and estimation of the effective diffusion coefficient in milk protein concentrate powder were carried out in accordance with the method described by Maidannyk et al., 2019 [4]. Rhodamine B without Polyethylene glycol was added to anhydrous nitrogen-injected MPC powder. Images were captured at short time intervals (approximately 3 s) using a Leica TCS SP5 CLSM, and particle diameters were measured using the Leica TCS SP5 software (Version 24.05.2023) in the size range 0.5–150 µm. The measuring was done manually, 5 to 7 times for each particle, and averaged result was used in the area calculations. The areas of single powder particles were calculated using spherical approximation. The time of the end of diffusion was estimated individually for each single particle. The effective diffusion coefficient was estimated as the ratio of the initial particle area to the full diffusion time (Equation (1)) [7]:(1)$$D_{eff}=\frac{Initial\;area\;of\;single\;particle}{Full\;time\;of\;diffusion}$$

### 2.4. Particle Size Distribution and Morphology

The particle size distribution of the MPC powder was determined using a Malvern Morphologi G3 (Malvern Instruments, Malvern, UK) with a sample dispersion unit (SDU) image analysis-based particle characterization system [15]. The sample (~10 mg) was placed in the sample entrainment spool and inserted into the top of the dispersion chamber. Then, the sample was dispersed with 1 Bar injection pressure on a 180 × 110 mm glass plate slide. A 10× objective and 80% aperture diagram level were used. The focus and light/threshold settings were manually checked and automatically calibrated with a target intensity of 80.00%. Size measurements were recorded as the median diameter D50 and cumulative diameters D90 and D10 whereby 50, 90 and 10% of the volume is smaller than the size indicated. The volume-weighted mean diameter (D (4,3)), elongation and circularity were also calculated.

### 2.5. Image Analysis through Matlab©

Image analysis was carried out by Matlab© (The Mathworks, Natick, MA, USA). Rehydration of each particle powder was represented by the penetration of the dye monitored by its fluorescence intensity. To characterize structural information and rehydration of the powder particle, a series of slices were primarily segmented into the background and rehydration region in the binary images based on a greyscale threshold using Otsu’s method [16]. Further, powder particles in the images were labeled from the binary images. By filling the holes in the area region covering by the rehydration of the powder particles of the binary images, the morphological properties of each powder particle, such as total area, diameter, centroids, boundary box, were computed and recorded. The full script is available in the Appendix A.

#### 2.5.1. Rehydration Analysis of Each Particle

The rehydration of each particle powder was represented by the penetration of the dye monitored by its fluorescence intensity. A gradient of fluorescence intensity representing a gradient of hydration was observed from the boundary to the center of the powder particle.

To calculate the level of penetration of the dye during rehydration, a relative intensity of the dye representing its concentration was calculated in each powder particle as follows (Equation (2)):(2)$${Int}_{rel}=\frac{Int-{Int}_{min}}{{Int}_{max}-{Int}_{min}}=I_{norm}$$
where *Int* is the fluorescence intensity of the dye in each powder, *Int_max_* and *Int_min_* are the maximal and the minimal fluorescence intensities of the dye in each powder particle, respectively, *Int_rel_* is the relative or normalized fluorescence intensity. The mean relative intensity for each powder particle as a function of time was computed to identify the rate of hydration process of the powder particle.

#### 2.5.2. Diffusion Model

Since the gradient of fluorescence intensity represents the gradient of hydration, the molar transport of the dye with respect to time assumed to be governed by second Fick’s law of diffusion due to the concentration gradient was described as follows (Equation (3)) [17]:(3)$$\frac{\partial {Int}_{rel}}{\partial t}=\nabla D_{eff}\nabla {Int}_{rel}$$
where *D_eff_* (m^2^/s) is the diffusion coefficient of the dye in the powder particle, *∇* (m^−1^) is the gradient operator, *t* (s) is the time. Equation (3) describes the accumulation rate of the dye concentration within the volume as proportional to the local change of the concentration gradient.

From the images obtained from a confocal laser scanning microscope, MPC powder particles were relatively spherical. Assuming the spherical geometry of the powder particle, relative quantity of the dye $I_{theor}(t)$ which entered the powder particle, the time *t* can be analytically expressed by Crank (Equation (4)) [18]:(4)$$I_{theor}(t)=\frac{{Int}_{rel}(t)}{{Int}_{rel}^{\infty }}=1-\frac{6}{\pi^{2}}\sum_{n=1}^{\infty }\frac{1}{n^{2}}\exp\left(-\frac{D_{eff}n^{2}\pi^{2}t}{R^{2}}\right)$$
where ${Int}_{rel}^{\infty }$ is the relative intensity of the dye in the solution surrounding the powder particle, *R* (m) is the radius of the particle, ${Int}_{rel}(t)$ is mean relative intensity of the dye at the time.

### 2.6. Image Analysis through Machine Learning Algorithms

The same series of micrographs were used in image analysis through machine learning algorithms. The procedure was programmed in Python 3.11. This approach included the following steps: neural network training, particle marking, determination of the moment of diffusion completion, estimation, and clarification of effective diffusion coefficient by Equation (1).

The comprehensive documentation accompanying the source code for image analysis procedures (Python) used in this study was prepared and available in the Appendix A. This documentation elucidates the specific algorithms and techniques implemented. The Appendix A include a thorough description of key algorithms, such as segmentation techniques, morphological operations, neural network architectures, training procedures and provides step-by-step explanations of the code, ensuring that readers and fellow researchers can understand and replicate the analysis process.

### 2.7. Data Analysis

Morphology analyses were performed in triplicate. Statistical analysis was performed using a paired-sample *t*-test using Microsoft Office Excel 2011 (Microsoft, Inc., Redmond, WA, USA). Means differ significantly from each other if *p* < 0.05 and very significantly if *p* < 0.01.

## 3. Results

### 3.1. Original CLSM Images and Estimation of Effective Diffusion Coefficient

#### 3.1.1. Particle Size Distribution, Elongation and Circularity of MPC Powder

The particle size distribution curve of anhydrous MPC powder is shown in Figure 1a. The diameter values range from 0.2 to 73.6 µm with (*D_mean_*) 29.4 µm. Powder had monomodal volume-based distribution. The majority of powder particles were in size 0–80 µm, which is in agreement with other spray-dried high-protein powders [7,19]. Powder particles (~10,000) were analyzed with respect to two shape factors—elongation and circularity. Figure 1b,c shows that the majority of MPC powder particles have a very high circularity and very small elongation and can roughly counted as spheric shape particles. The particle size distribution was used for diffusion coefficient estimation in all compared diffusion models.

#### 3.1.2. Original CLSM Micrographs and Effective Diffusion Coefficient Estimation

The set of original confocal laser scanning micrographs showing the penetration of the rhodamine B water dye into MPC powder particles is displayed in Figure 2. The first image was captured after 27 s of dye addition to MPC powder.

Diffusion of Rhodamine B molecules into the anhydrous MPC powder particles allowed for the visualization of water penetration into the particle center, indicative of powder hydration, while preventing particle swelling and any disruption to shape and structure. Complete dye penetration to the center of the powder particle occurred after approximately 6 min, which is in agreement with previous results for high-protein systems [4,7,19]. However, the time of penetration depends on the particle size. The effective diffusion coefficient was calculated based on the penetrated area (with a spherical approximation) and the complete time of penetration (Equation (1)). The number of particles analyzed manually is apparently small due to the hard determination of the ending point of diffusion and the difficult measurement of the penetrated area. The effective diffusion coefficient increased linearly with increasing particle size distribution (Figure 3). Knowing this linear dependence and particle size distribution allows an estimated average water diffusion coefficient for MPC powder, which was 6.40 × 10^−13^ m^2^/s.

### 3.2. Image Analysis through Matlab©

#### 3.2.1. Diffusion Coefficient Estimation

The parameter *D_eff_* of the dye representing hydration rate for each powder particle was estimated by minimizing the sum square difference between the mean relative intensity at different times predicted by the model (Equation (4)) and the measured ones using a nonlinear estimation program written in Matlab.

#### 3.2.2. Characterisation of Milk Particles during Rehydration

Concerning the microstructure characterization of analyzed samples, the images were segmented into the background and rehydration region in the binary images and shown in Figure 4.

The same with morphology observation, the shapes of powder particles were irregular but close to spherical, and numerous hollows could be observed in the 2D image, confirming the slow dehydration of MPC. Examination of Figure 4 shows that the local diffusion of the dye was heterogeneous and varied in different MPC particles. Powder particles in the images were labeled from the binary images. The areas of particles were filled, and the total area of each particle was calculated. The areas of particles ranged from 2.27 × 10^−11^ to 1.41 × 10^−9^ m^2^. An equivalent diameter was subsequently calculated from the area. The diameter of particles ranged from 5.37 to 42.4 μm with an average of 19.2 ± 11.0 μm.

#### 3.2.3. Local Diffusion of the Dye in MPC Particle

To characterize local diffusion on each labeled MPC particle, the boundary box of each labeled MPC particle was computed on each image recorded during the diffusion process. The computed bounding boxes specified for each labeled MPC particle allowed tracking and analyze local diffusion of the dye in the labeled MPC particle within the boundary boxes in the originally greyscale images. Penetration of the dye in a typical MPC particle within the computed boundary boxes as a function of time is shown in Figure 5.

The dye’s gradient of penetration was illustrated by the fluorescence intensity gradient. To calculate the level of penetration of the dye during rehydration in the MPC particle, the relative intensity of the dye was calculated within the computed bounding box of the original greyscale image (Equation (2)). Subsequently, the mean relative intensities for the labeled MPC particle at different time steps were computed. At the initial time, the average relative intensity was 0.41 ± 0.10, indicating the dye penetrated into the MPC particle to a certain extent. From the fluorescence intensity of the dye interacting with the protein of the MPC particle shown in Figure 6, diffusion of the dye followed the route network toward the center of the MPC particle, confirming that the MPC particle was not homogeneous.

Changes in relative intensity of fluorescence signal as a function of time were presented for the level of dye penetration in the MPC particle (Figure 7). The relative intensity increased nonlinearly during 340 s. At the end of the experiment, average relative intensity increased up to 0.77 ± 0.05.

#### 3.2.4. Diffusion Coefficients

To predict the diffusion coefficient of the dye into each MPC particle, the MPC particle was approximated by spherical geometry with an equivalent diameter computed in Section 3.2.3. The diffusion coefficient was estimated by fitting the solution of the diffusion equation (Equation (4)) to the measured relative intensity of each MPC particle. The fitted model is clearly described by the experimental data (Figure 8).

The diffusion coefficients of different MPC particles are shown in Table 1. The average diffusivity in an image set was 3.39 × 10^−14^ m^2^/s with an average diameter of 19.2 μm. The particle size distribution allows the calculation of an average effective diffusivity for all ranges of MPC powder and it was 4.83 × 10^−14^ m^2^/s.

In the next step, the correlation between particle size and the diffusion coefficient was determined (Figure 3). Increased particle size increased diffusion coefficient.

### 3.3. Image Analysis through Machine Learning

#### 3.3.1. Raw Data Preparation

The set of confocal microscopy micrographs, described in Section 3.1, was expanded by image augmentations (rotations, axis stretching, and a combination of these transformations). A subset of the images from the resultant data set underwent manual annotation via the service available at [https://www.makesense.ai, accessed on 1 November 2023]. The markup was carried out in accordance with the Coco format [https://cocodataset.org/, accessed on 1 November 2023].

#### 3.3.2. Neural Network Learning

The pretrained [maskrcnn_resnet50_fpn] was taken as the base neural network. To enhance the neural network’s capability to distinguish between powder particles and other elements in the micrographs, the architecture was modified by substituting the network’s original classifier with a new prediction head designed for binary classification.

The updated neural network underwent training for 280 epochs using a data set of 44 images, with a batch size equal to 4. Quality was checked on the validation data set. The achieved Intersection over Union (IoU) metric on the deferred (validation) stage was 90%. The weights of the resulting neural network were used to automatically select particles in one set.

#### 3.3.3. Particle Labeling

Each particle labeling was characterized by a set of features (coordinates of the corner of the selected square that includes the particle, the width of the square, and the height of the square). The most identical particles were selected using the Nearest Neighbours method in photographs adjacent in time. It allowed us to analyze the time series of each individual particle. The particle mask was defined for each individual particle in each micrograph. The minimum intensity—*int_min_* was obtained as the minimum intensity value for the initial micrograph, and the maximum intensity was taken as the maximum intensity—*int_max_* from the last photo in the series. The relative average (normalized) intensity was calculated by Equation (2).

The following values were also calculated:Quantile 1 (q1) of the intensities of all pixels of the particle;The intensity value of the pixels inside the particle (this intensity was normalized by the Min Max method, where the maximum value was equal to 255 (the maximum value for the picture), the minimum was taken from the initial micrograph (the minimum intensity value for all pixels);The standard deviation of normalized intensity.

#### 3.3.4. Determination of Diffusion Completion Moment

The termination point of the diffusion process was defined based on multiple criteria, incorporating empirically obtained constants that were expertly chosen during data analysis. The fulfillment of one of the following conditions is regarded as the moment of the end of diffusion: (I) The decrement of q1′s rate of change is less than −4.99 between consecutive micrographs, and q1 surpasses the empirical threshold of 80, equivalent to 31% of the pixel’s maximum intensity; (II) q1 exceeded the value of 100, denoting 40% of the maximum available intensity of one pixel; (III) The change in the standard deviation of the normalized intensity was less than 0.0001, the standard deviation of the normalized intensity fell below 0.18 and q1 was greater than 80.

For each particle, the moment of the end of diffusion was determined. If neither criterion was achieved, then the diffusion process was not considered completed. Figure 9 shows the change in quantile 1 and standard deviation of normalized intensity over time (the *x*-axis is the time in seconds from the beginning of the experiment).

#### 3.3.5. Estimation and Clarification of the Diffusion Coefficient

The diffusion coefficient was approximated by calculating the ratio of the initial area of the particle, captured at the first measurable instance, to the time at the termination point of diffusion (Equation (1)). Figure 3 shows the dependence of the diffusion coefficient on the particle diameter.

For clarification of the diffusion coefficient, Equations (1) and (3) were minimized and (*I_norm_* − *I_theor_*)^2^ values were optimized. Optimization was carried out using the Nelder–Mead simplex algorithm [20]. The previously obtained values of the diffusion coefficient (Figure 3) were used as an initial approximation. The average diffusivity for all powder particles was 1.39 × 10^−12^ m^2^/s with an average diameter of 15.2 μm.

## 4. Discussion

For all compared methods, the D_eff_ increased with increasing particle size of MPC powder (Figure 3). Similar results were also found in previous studies [4,5,6,7,19]. The microstructure of milk powders is significantly affected by their chemical composition and spray drying condition, method and geometry [21,22,23,24,25]. Small particles are usually compact with low porosity, representing a low diffusion coefficient. The large particle might contain hollow structures on its surface that allow capillary penetration of the dye solution in addition to hydrophilic interaction between water and protein. That can explain the large diffusion coefficient of large MPC particles. The hydrophilic interaction of MPC might be weaker than other hydrophilic compound such as lactose or minerals. Adding strongly hydrophilic compounds, such as lactose or minerals, can reduce the wetting time [21] and, hence, in turn, affect the functional properties of dairy powders.

Even though for all compared models, the trend in *D_eff_* vs. particle size is the same, they increase differently. Manually calculated and calculated by Python *D_eff_* showed linear dependence, with a much more dispersed result (R^2^ = 0.6665 vs. 0.9976 for manual) for Python calculation due to a higher number of investigated particles. At the same time, for the Matlab method, polynomial approximation in secondo rder provides a better fit than linear approximation (R^2^ = 0.9558 vs. 0.8842 for linear).

All compared methods show statistically the same results for average *D_eff_* values (Figure 10). The difference in results between the manual, Python and Matlab methods is associated with a different approach in determining the moment of the end of diffusion. In the manual and Python method, the time of completion of diffusion was considered the moment of penetration, while in the Matlab method, the time of diffusion was considered the moment of constant signal intensity. Similar average effective diffusivity values were published before for anhydrous spray-dried intact and hydrolyzed whey protein powders [26], amorphous spray-dried lactose [27], skim milk [28] and MPC powders [5]. Hence, all investigated methods are in agreement with literature data and may be applied for *D_eff_* calculations. However, all studied models have advantages and limitations.

The set of micrographs was made by CLSM. For this, liquid rhodamine B solution was added to the MPC powder, and the final dispersion was covered by a cover slide. Later, between the cover slide and the objective, a drop of immersion oil was placed. Hence, the powder particles continuously moved in the dispersion. This movement of particles was negligible and was not considered. Also, due to apparently low time of processing, the possible swelling of particles during rehydration was not considered. However, both processes are potential sources of error. Additionally, in this study, several important assumptions and limitations were made:Particle Sphericity. The calculations of the effective diffusion coefficient assume that the powder particles are spherical. However, particles may have irregular shapes, which could affect the accuracy of diffusion measurements.The same imaging conditions. The confocal microscopy images were captured under uniform settings, which is a limitation as it might not represent the variability encountered in practical scenarios. Sensitivity analyses could be conducted by capturing images under a range of conditions (e.g., varying magnifications, lighting, contrast settings, etc.) to assess the robustness of findings under diverse imaging conditions.The *D_eff_* of Rhodamine B dye water solution is the same as the *D_eff_* of water. Even though 0.1% (wt/wt) Rhodamine B has a slightly different *D_eff_* value [29], this difference was not considered in this work.Temperature Control. Although the experiments were conducted under the assumption of constant temperature, temperature fluctuations can significantly affect diffusion rates. Incorporating temperature controls in future experiments or conducting a sensitivity analysis to determine the impact of temperature fluctuations on diffusion rates would enhance the robustness of the results.

In particular, the manual measurement of effective diffusivity is very labor-intensive and requires a trained person. Diameter estimation is needed for each particle, which requires manual measurement of the average diameter for each particle at each time interval. This means that for a large number of particles, there are hundreds of manual measurements and subsequent calculations. Also, the estimation of particle diameter is not precise, which results in high errors and deviations. The number of investigated particles is significantly lower in comparison with other methods, which makes data statistically poor [30]. As an advantage of manual measurement is no need for special software and *D_eff_* may be simply estimated by various available photo redactors.

At the same time, Matlab image analysis also has some important limitations. First, the fluorescence intensity generated by the interaction between the dye and protein should be large enough to make a good distinction between the powder particles and the background during image segmentation [31]. Also, during sampling of particle rehydration, particles should be in relative separation from each other and not overlap to observe dye penetration on each particle. A high correlation was assumed between the fluorescence intensity of the dye and its concentration, representing water penetration into MPC particles. Furthermore, the diffusion coefficient estimated from Equation (3) was only implemented with approximately spherical geometry of powder particles. For irregular shapes of particles of the powder, the finite element method would be required to compute the diffusion coefficient. Nevertheless, this calculation method is built on the combination of CLSM, imaging analysis and diffusion model and can serve as an effective approach to analyze the penetration of water inside milk powder particles during the rehydration process.

The calculation of effective diffusivity through machine learning algorithms by Python has the following disadvantages and potential challenges. This method is data-dependent. The machine learning models were trained on a specific set of images, which may limit their generalizability to images with different characteristics. The quality of the hand-tagged images used for training the model is crucial. Poorly tagged images can lead to incorrect learning and subsequent analysis. Machine learning models heavily rely on the volume and quality of the training data, and any shortcomings in the data set can negatively impact the model’s performance [32]. There is always a risk that the model may become too tailored to the training data set, compromising its ability to perform well on new, unseen data. Addressing this issue will require the use of transfer learning techniques and the inclusion of a more diverse set of images in the training data set, covering a wider range of particle sizes, shapes, and imaging conditions. The training of machine learning models, particularly neural networks, requires significant computational power, which can be a constraint for some research environments. Accurately labeling a large set of images for training a machine learning model can be a highly intricate and time-consuming process that requires domain expertise [33]. Also, the choice of algorithm and its configuration, such as the decision to replace the head of the neural network with a two-class predictor, can influence the model’s performance [34]. The advantages of this model include high accuracy and precision. The machine learning model achieved a high Intersection over the Union (IoU) metric of 90%, indicating precise segmentation and identification of powder particles against the background. Automation and Speed: Once trained, machine learning algorithms can automatically analyze large data sets of images much faster than manual tagging, significantly reducing the time required for analysis. Reproducibility: Machine learning models can be run multiple times with the same data to obtain consistent results, which is essential for scientific studies. Dynamic Analysis: The use of machine learning allows for the dynamic analysis of particles over time, tracking changes and providing insights into the diffusion process at individual particle levels. Advanced Data Handling: Machine learning can handle complex patterns and variations in data that might be too subtle or intricate for traditional image analysis methods.

## 5. Conclusions

This study presents a detailed comparison of two independent image analysis methods for calculating the effective water diffusion coefficient of milk protein concentrate powder from the same set of confocal microscopy micrographs: conventional image analysis using Matlab and image analysis using Python machine learning algorithms. The results obtained by these methods were compared with manual measurements and previously published results for similar high-protein spray-dried systems. Both methods showed statistically equivalent results and can be used to estimate the effective diffusion coefficient of water. Despite some limitations, both methods show significant advantages over manual measurement. These automated calculation approaches can simplify the characterization and further optimization of the complex hydration process of high-protein systems. The findings from this study significantly improved the quality and speed of image analysis and may be used as a base for a simple and end-user-friendly software for powder hydration estimation, which is of practical interest for final powder properties and for stability and quality control during production and storage.

## Figures and Tables

**Figure 1 foods-13-00094-f001:**
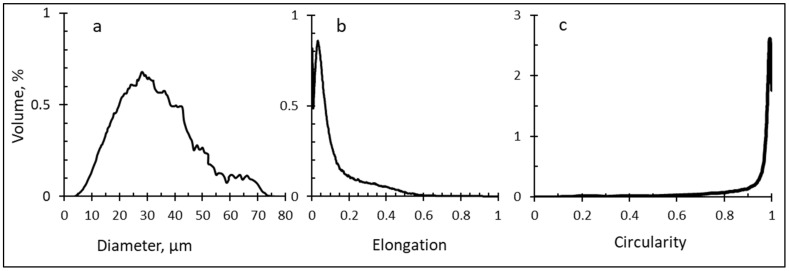
Anhydrous milk protein concentrate powder particle size distribution (**a**), elongation (**b**) and circularity (**c**).

**Figure 2 foods-13-00094-f002:**
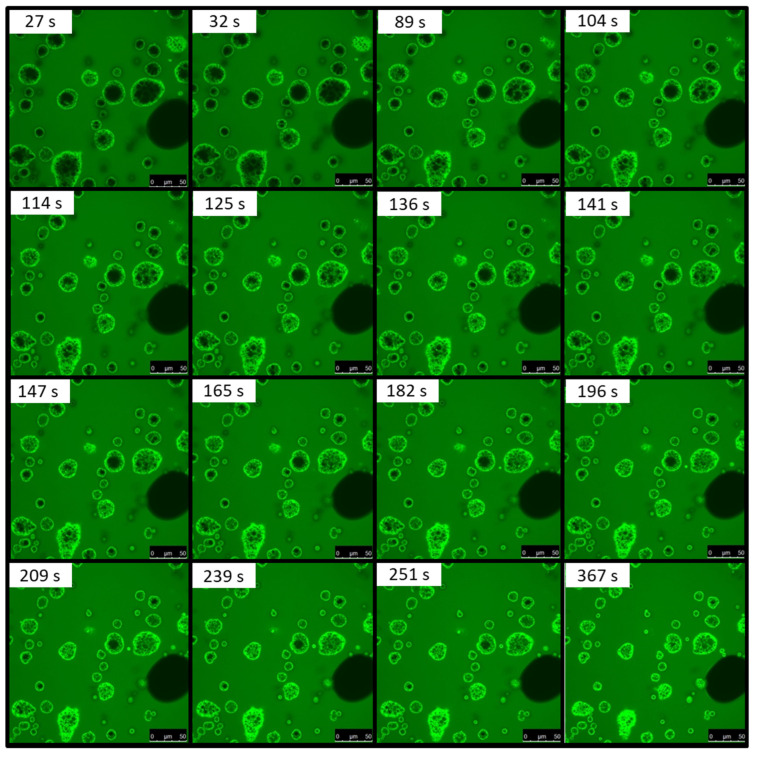
Confocal laser scanning microscopy images showing the movement of Rhodamine B dye into milk protein concentrate powder over time (s).

**Figure 3 foods-13-00094-f003:**
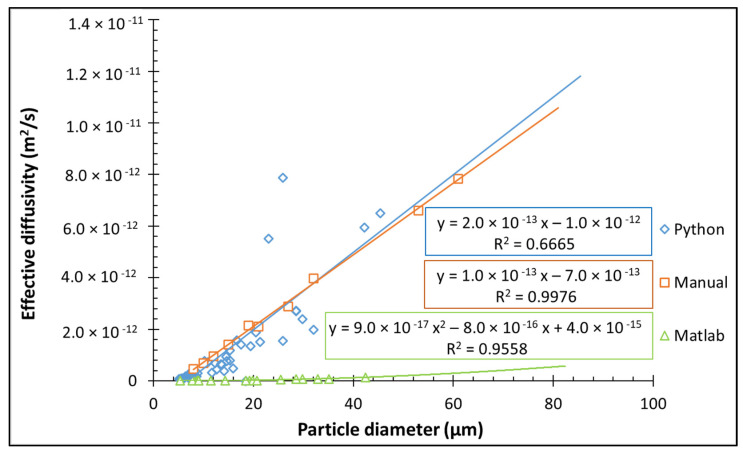
Effective diffusivity versus particle size through Python, Matlab and manual calculations.

**Figure 4 foods-13-00094-f004:**
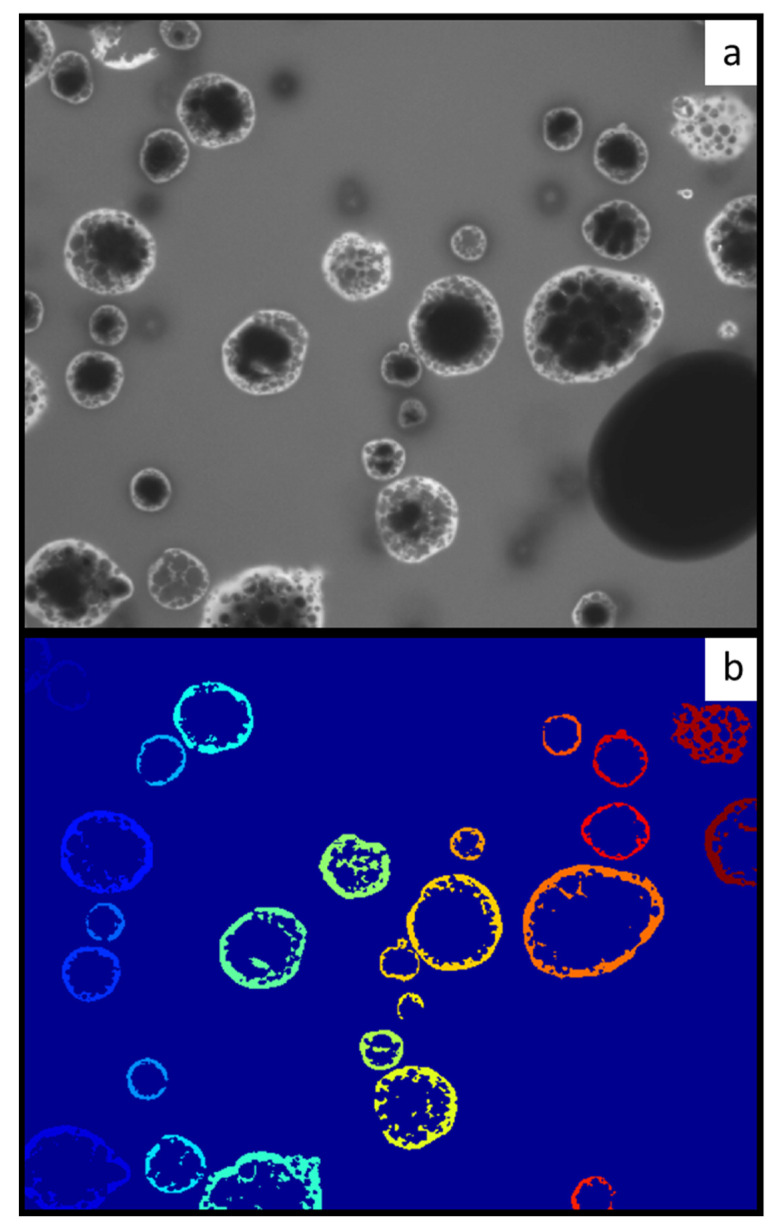
Microstructure of MPC (**a**) Initial grey scale image (**b**) after segmentation and labeling. Each colour represents particle with different diameter.

**Figure 5 foods-13-00094-f005:**
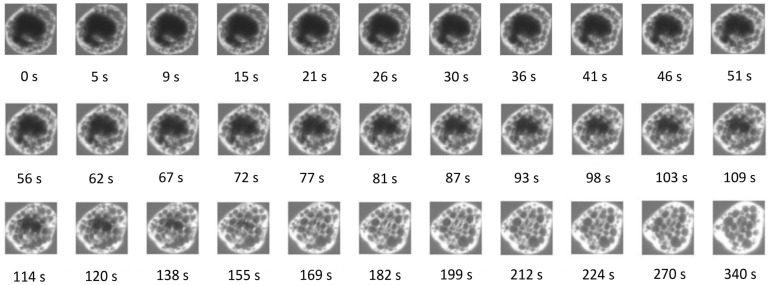
Fluorescence intensity due to penetration of the dye in a MPC particle within the computed boundary box during rehydration.

**Figure 6 foods-13-00094-f006:**
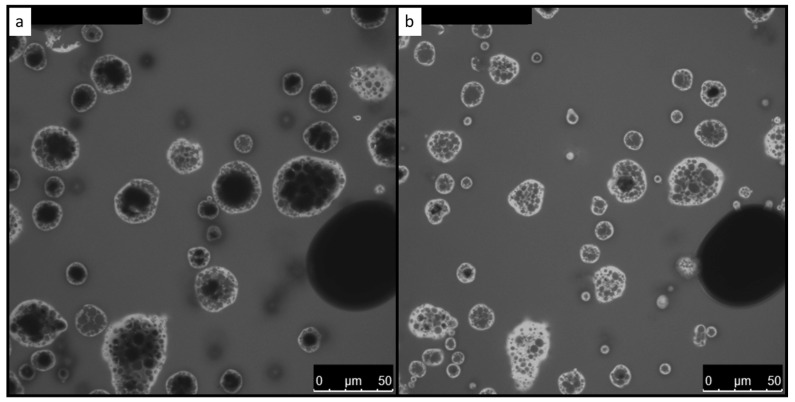
Fluorescence signal of the dye in MPC particles at the initial time (**a**) and after 340s (**b**).

**Figure 7 foods-13-00094-f007:**
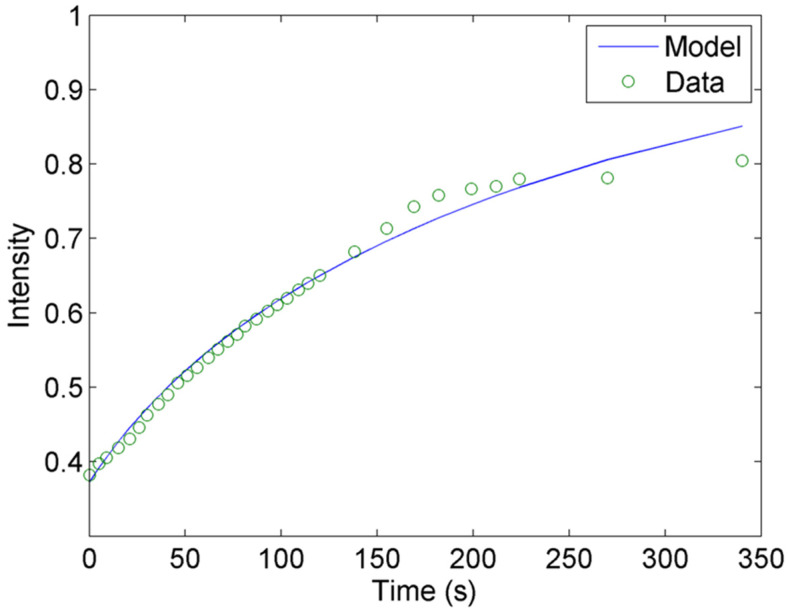
Change of relative fluorescence intensity in a typical MPC particle as a function of time due to interaction of the dye and protein during rehydration. The symbol indicates the measurement, while the line shows the fitted model.

**Figure 8 foods-13-00094-f008:**
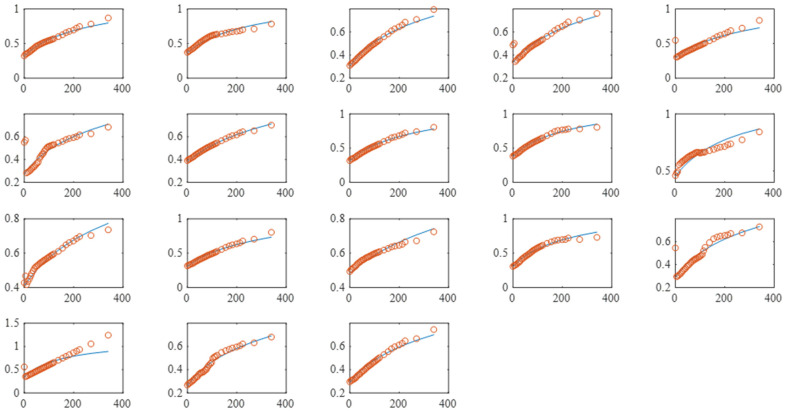
Change of relative fluorescence intensity in different MPC particles as a function of time during rehydration. Labels of *x* and *y* axes are time (s) and relative fluorescence intensity, respectively. The symbol indicates the measurement, while the line shows the fitted model.

**Figure 9 foods-13-00094-f009:**
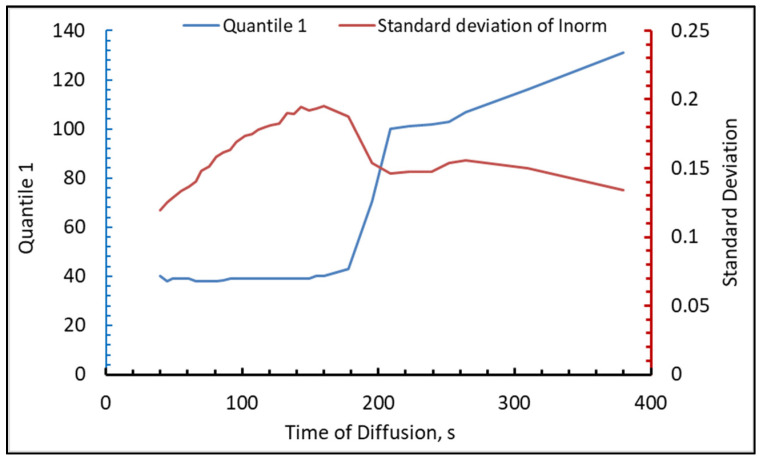
The example of Quantile 1 and Standard Deviation of normalized intensity versus time of diffusion for individual MPC particle with diameter 13.7 µm.

**Figure 10 foods-13-00094-f010:**
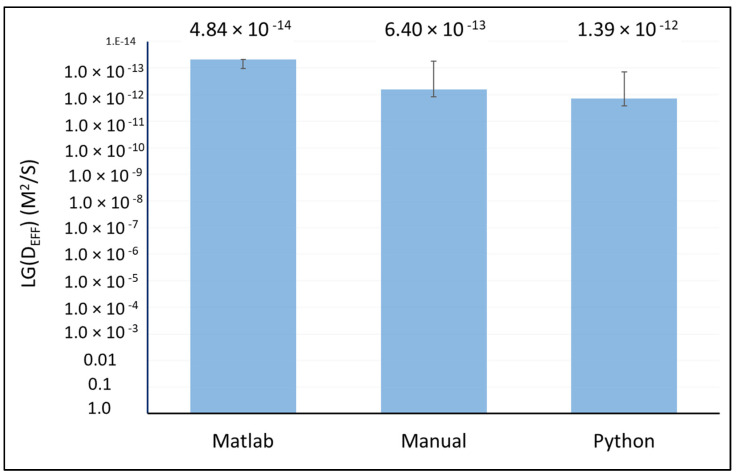
Average Effective Diffusivity *D_eff_* (m^2^/s) calculated using Matlab, manual and Python methods.

**Table 1 foods-13-00094-t001:** Morphological properties, relative intensity of the dye and estimated diffusion coefficients of MPC particles.

Particle ID	Area (μm^2^)	Equivalent Diameter (μm)	Relative Intensity at Initial Time	Relative Intensity at 340 s	Diffusion (m^2^/s)
1	850.13	32.90	0.324	0.869	8.25 × 10^−14^
2	696.65	29.78	0.372	0.784	7.00 × 10^−14^
3	290.77	19.24	0.310	0.795	2.13 × 10^−14^
4	52.01	8.14	0.487	0.761	3.67 × 10^−15^
5	59.40	8.70	0.547	0.835	4.14 × 10^−15^
6	63.33	8.98	0.551	0.684	4.14 × 10^−15^
7	311.11	19.90	0.391	0.702	1.81 × 10^−14^
8	509.43	25.47	0.318	0.805	4.50 × 10^−14^
9	638.87	28.52	0.382	0.805	7.63 × 10^−14^
10	162.72	14.39	0.461	0.842	2.00 × 10^−14^
11	22.65	5.37	0.427	0.736	1.76 × 10^−15^
12	969.63	35.14	0.314	0.804	6.88 × 10^−14^
13	104.01	11.51	0.493	0.724	5.94 × 10^−15^
14	1410.64	42.38	0.306	0.730	1.43 × 10^−13^
15	47.15	7.75	0.546	0.730	3.40 × 10^−15^
16	64.26	9.05	0.558	0.789	5.63 × 10^−15^
17	338.16	20.75	0.268	0.682	2.06 × 10^−14^
18	266.73	18.43	0.295	0.746	1.66 × 10^−14^

## Data Availability

Data is contained within the article or Appendix A.

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
