# Peer review of "Water Effective Diffusion Coefficient in Dairy Powder Calculated by Digital Image Processing and through Machine Learning Algorithms of CLSM Micrographs"

_foods, 2023, doi:10.3390/foods13010094_

Round 1

Reviewer 1 Report

Comments and Suggestions for Authors

The article titled "Water Effective Diffusion Coefficient in Dairy Powder calculated by Digital Image Processing and through Machine Learning Algorithms of CLSM Micrographs" presents a study on the rehydration process of dairy powders using the effective diffusion coefficient. The authors propose a modified method that combines confocal microscopy, image analysis techniques, and machine learning algorithms to measure the real-time water diffusion coefficient in dairy powders. Overall, the study is well-conducted and provides valuable insights into understanding the hydration process of high-protein powders. However, there are some areas that require clarification and improvement. Detailed comments are provided below.

The article is generally well-written and organized, with a clear structure and logical flow of information. The language is concise and appropriate. However, there are a few instances where clarification is needed, particularly in the Materials and Methods section.

The study addresses an important research gap in understanding the hydration process of high-protein powders. The combination of confocal microscopy, image analysis techniques, and machine learning algorithms to estimate the effective diffusion coefficient is novel and contributes to the existing body of knowledge. The findings have potential implications for the food and dairy industry, as well as pharmaceutical applications. Therefore, the study holds both scientific and practical significance.

The research methodology is generally sound and appropriate for achieving the stated objectives. However, there are some areas that require further clarification and improvement to ensure the reproducibility of the study.

Specific Comments:

The Methods section should provide more specific details on the acquisition parameters used during confocal microscopy, such as scanning resolution, time intervals, and laser power. These details are crucial for replication and validation of the results.

The mathematical formula used for calculating the effective diffusion coefficient should be explicitly stated, including any assumptions made. Providing a clear equation will enhance the transparency of the analysis.

The description of the image analysis procedures in Matlab and Python should be more detailed, including the specific algorithms and techniques utilized. This will allow readers to understand the steps involved in the analysis and potentially replicate the study.

The description of the particle size distribution analysis using the Malvern Morphologi G3 system should include the specific settings and parameters used. This information is essential for reproducibility and comparison with future studies.

The limitations of the proposed methodology should be discussed, including any potential sources of error or uncertainty. Addressing the limitations will strengthen the study and provide a more comprehensive understanding of the results.

Based on the above assessment, I recommend that the authors undertake major revisions to address the mentioned points. These revisions are necessary to ensure the scientific rigor, clarity, and reproducibility of the study.

Author Response

Dear reviewer,

Thank you for your time and review.

The manuscript was updated according to the comments and suggestions.

Please find attached file with our answers.

Sincerely,

Dr. Valentyn Maidannyk.

Reviewer 2 Report

Comments and Suggestions for Authors

Dear Author,

I found the subject matter of your work to be captivating and likely to engage a wide readership. However, I have a few minor suggestions to enhance the overall quality of your manuscript.

1. Firstly, please address the resolution of the figures to ensure clarity. 2. Additionally, I noticed that the second part of the discussion following Figure (10) lacks references to substantiate your findings. Kindly consider incorporating relevant citations for added credibility.

Methodology Improvements and Further Controls:

3. The authors should provide more details on the modifications made to the labor-intensive method, ensuring clarity on the enhancements introduced.

4. consider providing information on the robustness and generalizability of the machine learning models, such as validation techniques and performance metrics.

5. Discuss potential limitations of the methodology, including any assumptions made, and propose additional controls or sensitivity analyses to strengthen the validity of the findings.

Consistency of Conclusion:

6. The conclusions appear consistent with the evidence and arguments presented. However, the authors should elaborate on the implications of their findings and discuss any potential practical applications or future research directions suggested by the results.

Additional comments:

1. It would be beneficial to include recent and pertinent literature to demonstrate awareness of the latest developments in the field.

2. Consider the layout and clarity of figures to enhance their visual interpretation.

3. Check for consistency in the use of units and scales across tables and figures.

Overall, the manuscript is promising, but further clarification and additional details in certain areas would enhance the robustness and clarity of the research.

Best regards,

Author Response

Dear reviewer,

Thank you for your time and review.

The manuscript was updated according to the comments and suggestions.

The article titled "Water Effective Diffusion Coefficient in Dairy Powder calculated by Digital Image Processing and through Machine Learning Algorithms of CLSM Micrographs" presents a study on the rehydration process of dairy powders using the effective diffusion coefficient. The authors propose a modified method that combines confocal microscopy, image analysis techniques, and machine learning algorithms to measure the real-time water diffusion coefficient in dairy powders. Overall, the study is well-conducted and provides valuable insights into understanding the hydration process of high-protein powders. However, there are some areas that require clarification and improvement. Detailed comments are provided below.
The article is generally well-written and organized, with a clear structure and logical flow of information. The language is concise and appropriate. However, there are a few instances where clarification is needed, particularly in the Materials and Methods section.
The study addresses an important research gap in understanding the hydration process of high-protein powders. The combination of confocal microscopy, image analysis techniques, and machine learning algorithms to estimate the effective diffusion coefficient is novel and contributes to the existing body of knowledge. The findings have potential implications for the food and dairy industry, as well as pharmaceutical applications. Therefore, the study holds both scientific and practical significance.
The research methodology is generally sound and appropriate for achieving the stated objectives. However, there are some areas that require further clarification and improvement to ensure the reproducibility of the study.
Specific Comments:
The Methods section should provide more specific details on the acquisition parameters used during confocal microscopy, such as scanning resolution, time intervals, and laser power. These details are crucial for replication and validation of the results.
The mathematical formula used for calculating the effective diffusion coefficient should be explicitly stated, including any assumptions made. Providing a clear equation will enhance the transparency of the analysis.
The description of the image analysis procedures in Matlab and Python should be more detailed, including the specific algorithms and techniques utilized. This will allow readers to understand the steps involved in the analysis and potentially replicate the study.
The description of the particle size distribution analysis using the Malvern Morphologi G3 system should include the specific settings and parameters used. This information is essential for reproducibility and comparison with future studies.
The limitations of the proposed methodology should be discussed, including any potential sources of error or uncertainty. Addressing the limitations will strengthen the study and provide a more comprehensive understanding of the results.
Based on the above assessment, I recommend that the authors undertake major revisions to address the mentioned points. These revisions are necessary to ensure the scientific rigor, clarity, and reproducibility of the study.
Dear reviewer, thank you very much for the reviewing of our manuscript and for comments, questions, and suggestions.
Please find our answers below:
Q: The Methods section should provide more specific details on the acquisition parameters used during confocal microscopy, such as scanning resolution, time intervals, and laser power. These details are crucial for replication and validation of the results.
A: The manuscript was updated. Please check lines: 78-109.
Q: The mathematical formula used for calculating the effective diffusion coefficient should be explicitly stated, including any assumptions made. Providing a clear equation will enhance the transparency of the analysis.
A: Since the gradient of fluorescence intensity represents gradient of hydration, the molar transport of the dye with respect to time is assumed to be governed by second Fick’s law of diffusion due to the concentration gradient. Eq. (2) describes the accumulation rate of the dye concentration within the volume as proportional to the local change of the concentration gradient.
Of note, for a complicated geometry, we can not obtain an analytical solution for Fick’s second law. In such case, numerical analysis is used. However, from the images obtained from confocal laser scanning microscope, MPC powder particles were relatively spherical. Therefore, quantity of the dye which entered the powder particle the time t can be analytical expressed by Crank [18].
Section 2.5.2 has been updated according to the suggestion of the review.
Q: The description of the image analysis procedures in Matlab and Python should be more detailed, including the specific algorithms and techniques utilized. This will allow readers to understand the steps involved in the analysis and potentially replicate the study.
A: The manuscript was updated, please see lines: 125-191.
Q: The description of the particle size distribution analysis using the Malvern Morphologi G3 system should include the specific settings and parameters used. This information is essential for reproducibility and comparison with future studies.
A: The manuscript was updated, please see lines: 111-123.
Q: The limitations of the proposed methodology should be discussed, including any potential sources of error or uncertainty. Addressing the limitations will strengthen the study and provide a more comprehensive understanding of the results.
A: The manuscript was updated, please see lines: 449-525.

Reviewer 3 Report

Comments and Suggestions for Authors

The manuscript entitled "Water Effective Diffusion Coefficient in Dairy Powder calculated by Digital Image Processing and through Machine Learning Algorithms of CLSM Micrographs is a well-written manuscript, nio major issue/issues detected in the text and can be furthere considered. The similarty percentage verified by Turnitin is also in the acceptable range of 16%. 

Thank you

Author Response

Dear reviewer,

Thank you very much for your time and review.

The manuscript was updated according to the comments.

We are appreciated!

Sincerely,

Dr. Valentyn Maidannyk.